# Non-Contact Measurement and Identification Method of Large Flexible Space Structures in Low Characteristic Scenes

**DOI:** 10.3390/s23041878

**Published:** 2023-02-07

**Authors:** Tianming Cheng, Xiaolei Jiao, Zeming Zhang, Qiang Bi, Cheng Wei

**Affiliations:** 1School of Astronautics, Harbin Institute of Technology, Harbin 150001, China; 2School of Aeronautics and Astronautics, Sun Yat-Sen University, Guangzhou 510275, China; 3Beijing Institute of Control Engineering, Beijing 100190, China

**Keywords:** feature point matching, large flexible space structures, non-contact measurement, modal parameter identification, low characteristic scenes

## Abstract

The end-operation accuracy of the satellite-borne robotic arm is closely related to the satellite attitude control accuracy, and the influence of the vibration of the satellite’s flexural structure on the satellite attitude control is not negligible. Therefore, a stable and reliable vibration frequency identification method of the satellite flexural structure is needed. Different from the traditional non-contact measurement and identification methods of large flexible space structures based on marker points or edge corner points, the condition of non-marker points relying on texture features can identify more feature points, but there are problems such as low recognition and poor matching of features. Given this, the concept of ‘the comprehensive matching parameter’ of scenes is proposed to describe the scene characteristics of non-contact optical measurement from the two dimensions of recognition and matching. The basic connotation and evaluation index of the concept are also given in the paper. Guided by this theory, the recognition accuracy and matching uniqueness of features can be improved by means of equivalent spatial transformation and novel relative position relationship descriptor. The above problems in non-contact measurement technology can be solved only through algorithm improvement without adding hardware devices. On this basis, the Eigensystem Realization Algorithm (ERA) method is used to obtain the modal parameters of the large flexible space structure. Finally, the effectiveness and superiority of the proposed method are verified by mathematical simulation and ground testing.

## 1. Introduction

With the development of space technology, spacecraft and their accessory parts—such as solar wings, truss brackets, antennas, and robotic arms—are increasingly presented in the form of large flexible structures [1]. To reduce the spacecraft’s load, the flexible parts are mostly made of lightweight and ultra-thin composite materials. This large deflection flexible material has the characteristics of low stiffness, low-frequency modal density, and relatively small modal damping ratio. When the satellite is on orbit, flexible attachments of the satellite are susceptible to external intensity, causing continuous vibration, which is difficult to attenuate. This will lead to changes in the overall dynamics model of the satellite, thus causing a great impact on the satellite attitude control [2], which will lead to the degradation and failure of the satellite performance [3]. Finally, the vibration will indirectly affect the accuracy of the end-effector of the on-board robotic arm. Therefore, many countries are studying the on-orbit identification of dynamic characteristics of flexible structures attached to spacecraft [4]. Due to the difference between the space environment and the ground environment, the vibration characteristics of the flexible structure are also different. The simulation often cannot effectively simulate the space environment, nor can it predict accidental external excitation. The simulation results cannot reflect the on-orbit state of the spacecraft. Therefore, the on-orbit identification of the vibration information of the satellite flexible appendage is urgently needed.

The existing on-orbit modal identification methods are mainly divided into two categories. One is based on the existing measurement devices or control system data on the star body to complete the measurement. For example, Tang et al. [1] proposed a method for on-orbit modal identification of flexible structures of spacecraft based on reaction wheel actuators. The other is to observe the vibration of large flexible structures by adding special measuring devices. This method is more accurate and reliable and can be divided into contact and non-contact [5]. In Japan, several experiments were conducted on the Engineering Test Satellite-6 and Engineering Test Satellite-8. The modal parameters of the solar wing are identified based on the data from the satellite’s own attitude control system and the data measured by the acceleration sensors attached to the flexible solar wing at the same time [6,7]. This contact measurement method attaches the measurement device to the surface of the flexible structure, which affects the dynamics information of the flexible attachment itself and leads to errors in the identification information. At present, the better solution is non-contact measurement. Non-contact optical measurement based on binocular cameras is an effective technical approach, which can be subdivided into optical measurements with and without marking points. Some international research institutions have explored the measurement method of unmarked points. The Image Science and Analysis Group (ISAG) performed modal identification of the ISS 2A solar wing by binocular camera [8], but only two feature points at the end of the 2A solar wing could be identified. NASA obtained more dynamic information about feature points in the PASDE experiment with an optical measurement scheme, but six cameras were used for this purpose [9], which increased the complexity of the hardware system and algorithm. Relevant research has also been carried out in China, but the main examples of research are focused on the method with marker points. Wu et al. [10] proposed a visual measurement method based on circular marker points. Zang et al. [11] also used the method of attaching circular reflective markers on the solar wing for modal identification. Qiu et al. [5] used self-luminous light sources as measurement points, and then used a binocular camera for parameter identification. All these on-orbit measurement schemes add other physical devices, such as passive reflective marker points or active light sources, to the binocular vision system. The introduction of physical devices increases the complexity of on-orbit measurement and reduces the stability of the measurement system. Therefore, studies regarding a modal measurement identification method with rich identifiable features and good stability without marker points are urgently needed.

However, there are certain technical difficulties in this kind of measurement method without marking points. If the corner points of the edge of the flexible sails are used as identification points, there are fewer identifiable features, which is not conducive to improving the identification accuracy. If the texture features—such as the grid corner points—of the solar wing are identified, different feature points are highly similar to each other. When the optical camera identifies the feature points and matches the feature points, the similar descriptors will lead to a higher mismatch rate, so that the spatial three-dimensional coordinates of the feature points cannot be accurately measured. Moreover, in the case of no identification points, the feature points extracted by the optical camera are often different between frames. Therefore, it is more difficult to obtain the continuous dynamic coordinate values of the feature points at fixed positions on the flexible solar wing compared to the case with identification points. In addition, the measurement accuracy of an optical camera is usually lower than that of traditional contact measurements, so the improvement of measurement accuracy is also an issue to be considered.

To solve the above problems, this paper first introduces the concept of the comprehensive matching parameter of scenes and then proposes a new feature point detection and matching method for the grid features on satellite flexible solar wings. This method improves the characteristic of the scene and the uniqueness of feature matching through mathematical methods such as projection transform and image processing methods such as clustering and binaryzation, as well as proposing new descriptors to solve the problems that features of the solar wing are highly repetitive and not easy to match. Finally, under the condition of no additional physical target, the spatial 3D coordinates of the grid feature points on the satellite flexible solar wings are continuously tracked by the binocular vision system and used for modal recognition. The on-orbit vibration recognition simulation test is completed for the low feature degree scene of the satellite flexible attachment, and the accuracy of the test results is relatively high, which verifies the feasibility of the theory in this paper.

## 2. Structural Dynamics Modeling Analysis of Flexible Solar Wing

In this paper, we construct a multi-body dynamics model [12] of the whole star based on the coupled satellite model of the central rigid body and flexible attachment shown in Figure 1, in which the central rigid body is C, the origin of the coordinate system, $O_{C}$ is located at the center of mass of the rigid body, and the two flexible solar wings are $Flex_{1}$ and $Flex_{2}$, respectively. The two flexible solar wings are fixed on the side near the satellite body and the ends are free. According to the hypothetical modal method, the elastic displacement of any point on the flexible solar wing is shown in Equation (1):(1)$$q(r,t)=\sum_{I=1}^{\infty }\varphi_{I}(r)\eta_{I}(t)=\Phi (r)\eta (t)$$
where Φ(r) is 3×∞ order modal function matrix and η(t) is ∞×1 order modal coordinate vector. According to the Lagrange equation, the dynamics equation of the system can be established as follows:(2)$$\begin{array}{c} m{\ddot{\omega }}_{0}(t)+P_{1}{\ddot{\eta }}_{1}+P_{2}{\ddot{\eta }}_{2}=\bar{F},\\ J\ddot{\theta }+H_{1}{\ddot{\eta }}_{1}+H_{2}{\ddot{h}}_{2}=\bar{T},\\ {\ddot{\eta }}_{1}+\Omega_{1}^{2}\eta_{1}+P_{1}^{T}{\ddot{\omega }}_{0}(t)+H_{1}^{T}\ddot{\theta }=f_{1},\\ {\ddot{\eta }}_{2}+\Omega_{2}^{2}\eta_{2}+P_{2}^{T}{\ddot{\omega }}_{0}(t)+H_{2}^{T}\ddot{\theta }=f_{2} \end{array}$$
where m is the total system mass, J is the system rotational inertia, $\omega_{0}$ is the translational displacement, θ is the rotational displacement, $\eta_{i}\left(i=1,2\right)$ is the flexible solar wing modal coordinate vector, $P_{i}$ and $H_{i}$(i=1,2) are the coefficient matrices of the modal momentum and angular momentum of the flexible solar wing, $\Omega_{i}\left(i=1,2\right)$ is the diagonal matrix whose diagonal parameters are the intrinsic frequencies of the flexible solar wing,  F ¯ is the external system force,  T ¯ is the external moment, and $f_{i}\left(i=1,2\right)$ is the flexible solar wing modal force. The system external force in this study is 0, i.e., $f_{1}=f_{2}=0$. The gyroscopic moment due to the maneuvering and damping of the flexible solar wing plane is $-\dot{\theta }\times J\dot{\theta }$. Neglecting the system advection, Equation (2) can be written as follows:(3)$$\begin{array}{l} J\ddot{\theta }+\tilde{\dot{\theta }}J\dot{\theta }+H_{1}{\ddot{\eta }}_{1}+H_{2}{\ddot{\eta }}_{2}=\bar{T},\\ {\ddot{\eta }}_{1}+2\xi_{1}\Omega_{1}{\dot{\eta }}_{1}+\Omega_{1}^{2}\eta_{1}+H_{1}^{T}\ddot{\theta }=0,\\ {\ddot{\eta }}_{2}+2\xi_{2}\Omega_{2}{\dot{\eta }}_{2}+\Omega_{2}^{2}\eta_{2}+H_{2}^{T}\ddot{\theta }=0 \end{array}$$
where $\xi_{i}\left(i=1,2\right)$ is the damping ratio of the flexible sail solar wing. Defining the generalized coordinate vector $X={\left[\theta^{T},\eta_{1}^{T},\eta_{2}^{T}\right]}^{T}$, then Equation (3) can be written in the form of Equation (4).
(4)$$M\ddot{X}+D_{p}\dot{X}+\mathrm{KX}=Q$$
where $M=\left[\begin{array}{ccc} J & H_{1} & H_{2}\\ H_{1}^{T} & I & 0\\ H_{2}^{T} & 0 & I \end{array}\right]$, $D_{p}=\left[\begin{array}{ccc} \tilde{\dot{\theta }}J & 0 & 0\\ 0 & 2\xi_{1}\Omega_{1} & 0\\ 0 & 0 & 2\xi_{2}\Omega_{2} \end{array}\right]$, $K=\left[\begin{array}{ccc} 0 & 0 & 0\\ 0 & \Omega_{1}^{2} & 0\\ 0 & 0 & \Omega_{2}^{2} \end{array}\right]$, $Q=\left[\begin{array}{c} \bar{T}\\ 0\\ 0 \end{array}\right]$.

Considering the output equation, the state space equation of the system can be written as follows:(5)$$\{\begin{array}{c} {\dot{x}}_{s}=A_{s}x_{s}+B_{s}u_{s}\\ y_{s}=C_{s}{\dot{x}}_{s}+D_{s}x_{s} \end{array}$$
where $x_{s}=\left[\begin{array}{c} \dot{X}\\ X \end{array}\right]$, $A_{s}=\left[\begin{array}{cc} -M^{-1}D_{p} & -M^{-1}K\\ I & 0 \end{array}\right]$, $B_{s}u_{s}=\left[\begin{array}{c} M^{-1}Q\\ 0 \end{array}\right]$, $C_{s}$ and $D_{s}$ are the observation matrices of velocity and displacement, respectively. $A_{s}$ and $B_{s}$ are constant coefficient matrices, determined by the parameters of the system itself, reflecting the characteristics of the system.

In this section, the dynamics model of the rigid-flexible coupled spacecraft is established, and the state–space equation of the system is obtained. This dynamic model can provide the output data of the system. We design the measurement scheme based on the dynamic model in this section to obtain the observed data for the identification of the vibration modal information of the flexible attachment.

## 3. On Orbit Measurement Scheme Design and Test Process

Figure 2 is the on-orbit measurement simulation test scheme used in this paper. The flexible solar wing coordinate system and camera coordinate system are shown in the figure. The test device simulates the satellite body and the unilateral solar wing. The flexible solar wing mainly vibrates freely along the $Z_{B}$ direction of its coordinate system. In the process of camera selection, the camera frame rate is considered to be 5 to 10 times of the sampling frequency. The camera resolution needs to ensure that the measurement accuracy reaches the submillimeter level. The global shutter camera is used, and the external signal generator controls the camera to trigger synchronously by outputting a fixed frequency square wave signal. Appropriate cameras are arranged as far as possible on one edge of the satellite center rigid body to increase the baseline length. The longer baseline length is beneficial to improve the measurement accuracy of the binocular system. The angle between the axes of the two cameras and the baseline is acute to increase the overlapping range of the two cameras’ field of view. Considering the camera image quality, the camera focal length is controlled in the middle of the flexible solar wing, thus ensuring the clarity of the acquired images. After the cameras take pictures, the image data are transmitted to the processor for post-processing through the network interface, and the whole system is powered by a DC power supply.

After setting up the hardware platform, the experiment is carried out according to the process shown in Figure 3. After the camera is installed, the calibration plate information is first collected to calibrate the camera, and the internal and external parameters of the camera are obtained. The calibrated camera is used to collect the vibration information of the flexible solar wing. The image pixel grayscale data are used as the input information of the feature enhancement and feature extraction module to detect the texture boundary of the solar wing. The feature points are identified, and the pixel coordinates of the feature points are used as output. This process is mainly completed under the guidance of the theory in Section 4. Then, pixel coordinates of the feature point are used as the input of the feature point spatial 3D information extraction module to calculate the spatial 3D position and velocity information of the target features. Finally, the three-axis velocity of the target feature point is input into the modal identification algorithm in Section 5 for modal analysis. The information of modal order, fundamental frequency, and modal coordinates are output. The related test equipment parameters, assembly relationships, and test results are presented in Section 6.

## 4. Feature Point Identification and Measurement Based on the Comprehensive Matching Parameter

### 4.1. The Concept and Connotation of Comprehensive Matching Parameter

In the experimental process shown in Figure 2, feature point extraction and matching are the key steps. In the existing optical measurement schemes with marker points, the presence of marker points directly provides identifiable features at fixed locations on the solar wing. It is only necessary to find the marker points in each image frame. The significant difference between marker points and background information greatly reduces the difficulty of feature extraction. Moreover, the marker points are clearly distributed. It is easy to find the correspondence of the marker points in the left and right eye images of the binocular camera, so the problems of feature point matching and 3D spatial information measurement are also easily solved. However, in the optical measurement problem without marking points studied in this paper, the camera does not have a clear identification target, and the corner points or speckle features of the solar wing grids are highly similar, so it is difficult to establish the correspondence between the same feature points in the left and right eye images, which is prone to mismatching. In addition, in the low-feature scene studied in this paper, the solar wing grid texture features are not easy to extract stably during its vibration process, and the features are highly repetitive. When dealing with such issues, many traditional algorithms can be used, such as DOG operator, SIFT operator, SURF operator, ORB operator [13,14,15], and so on. However, these methods ignore the enhancement of the scene features themselves and cannot improve the distinguishability of the scene features. Therefore, even if the binocular camera can correctly extract feature points, it is difficult to achieve the correct matching between the left and right eye images. Due to the lack of theoretical guidance, the existing unmarked point feature extraction and matching methods usually have no ideal practical effect. To solve this problem and complete the on-orbit measurement scheme, this paper proposes the following concepts.

**Definition** **1.**
*The comprehensive matching parameter of the scene is an index to judge whether the features of the scene itself are easy to identify and distinguish, and whether the feature point matching is robust and correct. This index is expressed by the letter ‘CMP’. It contains two contents: the characteristic of the scene and the uniqueness of the matching.*


**Definition** **2.**
*The characteristic of the scene refers to the property that feature points in the scene can be easily identified continuously and stably, and feature points have clear distinguishable indicators from each other. It is expressed by index ‘*

char

*’.*


**Definition** **3.**
*The uniqueness of matching means that the matching process of binocular camera feature points can form a one-to-one mapping relationship, i.e., it is always possible to find a unique and correct corresponding feature point in the right camera image for the feature point in the left camera image. It is expressed by index ‘*

MU

*’.*


**Definition** **4.**
*The relative position relation descriptor is a descriptor that describes the position-related features of feature points, which is used to distinguish repetitive features in low-feature scenes.*


This chapter will introduce the relevant content and usage of these definitions in detail. The Definitions 1–3 are used to solve the problem of optical measurement without marker points. Definitions 3 and 4 are used to solve the problem of how to extract the dynamic coordinates of feature points in time domain in different images without marker points.

### 4.2. Feature Enhancement and Extraction for the Low Comprehensive Matching Parameter Scenes

#### 4.2.1. Stable Extraction of Repetitive Features

As shown in Figure 4, the existing feature point extraction algorithms cannot guarantee that the feature points detected in one frame of the camera can still be detected in the next frame due to factors such as viewing angle, illumination, and resolution. However, the method in this paper requires stable tracking of fixed feature points in each frame to ensure stable extraction and stable matching. In this way, we can obtain continuous feature point position coordinate data and use them as input information for modal recognition.

To achieve stable feature extraction, the scene images are first clustered using the *K*-mean algorithm. The criterion function of the algorithm when there are *K* pattern classes is as follows:(6)$$J=\sum_{j=1}^{K}\sum_{i=1}^{N_{j}}{\left\|X_{i}-Z_{j}\right\|}^{2},\;X_{i}\in S_{j}$$
where $S_{j}$ is the j-th aggregation class and the clustering center is $Z_{j}$. $N_{j}$ is the number of samples contained in the j-th aggregation class $S_{j}$. According to the clustering algorithm criterion of the K-means algorithm, the clustering centers should be chosen such that the value of the criterion function J is extremely small, i.e.,
(7)$$\frac{\partial J_{j}}{\partial Z_{j}}=0$$
(8)$$\frac{\partial }{\partial Z_{j}}\sum_{i=1}^{N_{j}}{\left\|X_{i}-Z_{j}\right\|}^{2}=\frac{\partial }{\partial Z_{j}}\sum_{i=1}^{N_{j}}{\left(X_{i}-Z_{j}\right)}^{T}\left(X_{i}-Z_{j}\right)=0$$

Equation (8) can be solved as follows:(9)$$Z_{j}=\frac{1}{N_{j}}\sum_{i=1}^{N_{j}}X_{i},\;X_{i}\in S_{j}$$

For the satellite on-orbit scene, the K value of the pattern class is taken as 3, and the scene image is binarized according to the clustering results to highlight the desired features. Then, the Sobel operators $s_{x}$ and $s_{y}$ are introduced in the x and y directions of the image pixel coordinates, respectively, from which the gradient amplitude and direction at each pixel point of the image are calculated in turn as shown in Equation (10) and Equation (11), respectively.
(10)$$M\left(x,y\right)=\sqrt{{\left(\frac{\partial f\left(x,y\right)}{\partial x}\right)}^{2}+{\left(\frac{\partial f\left(x,y\right)}{\partial y}\right)}^{2}}$$
(11)$$\theta =\tan^{-1}\left[\frac{\partial f\left(x,y\right)}{\partial y}/\frac{\partial f\left(x,y\right)}{\partial x}\right]$$

When the gradient value is large, the corresponding pixel point is the boundary point on the image, so that the pixel points surrounding each grid of the flexible solar wing can be found. Pixel points $\left(x_{p}\left(i\right),y_{p}\left(j\right)\right)$ of a grid boundary constitute a set B, and the number of all pixels in the set is $size_{B}$. From this, the coordinates of the grid center feature points can be obtained as shown in Equation (12). Each grid center point can be found as a feature point. This method ensures the stability of feature point extraction and ensures that the found feature points can be evenly distributed on the surface of the solar wing, avoiding the aggregation of feature points.
(12)$$p_{feature}=\left(\frac{\sum_{i=1}^{size_{B}}x_{p}\left(i\right)}{size_{B}},\frac{\sum_{j=1}^{size_{B}}y_{p}\left(j\right)}{size_{B}}\right)$$

#### 4.2.2. Characteristic Enhancement

Scene feature points themselves have strong repeatability, that is, the ‘appearances’ of the feature points are similar, which is the inherent characteristic of the scene and cannot be changed. Therefore, this paper considers adding a unique ‘label’ to the extracted feature points from other angles as a distinction to enhance the scene characteristics. In the solar wing coordinate system shown in Figure 2, each detectable feature point in the solar wing plane is neatly arranged in the form of a “queue”. If the solar wing coordinate system plane $X_{B}O_{B}Y_{B}$ is transformed to be parallel to the camera pixel plane, the pixel coordinates of each feature point will also be neatly arranged in the form of “queue” and easy to distinguish. This transformation relationship is expressed by Equation (13), where $P_{N}$ is the pixel coordinates after projection transformation, $P_{P}$ is the pixel coordinates of the detected feature points in the original image, R and T are the rotation and translation transformation matrices between the camera coordinate system and the solar wing coordinate system, respectively. The equivalent transformation is shown in Figure 5, in which the original tilted observation view is transformed to a forward view perpendicular to the solar wing plane $X_{B}O_{B}Y_{B}$. In this process, the optical axis direction of the camera $u=u_{1}e_{1}+u_{2}e_{2}+u_{3}e_{3}$ tends to coincide with the normal vector of the flexible solar wing plane $v=v_{1}e_{1}+v_{2}e_{2}+v_{3}e_{3}$, i.e., Equation (14).
(13)$$\left[\begin{array}{c} X_{N}\\ Y_{N}\\ 1 \end{array}\right]=P_{N}=RP_{P}+T=\left[\begin{array}{ccc} r_{11} & r_{12} & t_{x}\\ r_{21} & r_{22} & t_{y}\\ 0 & 0 & 1 \end{array}\right]\left[\begin{array}{c} X_{P}\\ Y_{P}\\ 1 \end{array}\right]$$
(14)u⋅v→0

From Equation (14), it can be seen that this conversion process is only a change in the vector angle and does not change the nature of the feature points themselves. However, as shown in Figure 6, the figure illustrates the distribution of some feature points in one frame during the image acquisition. After characteristic enhancement, the feature points are neatly distributed in the pixel coordinate system with strong positional properties, and the feature point distribution tends to be more discrete, so that the originally chaotic and similar feature points are significantly distinguished. This means that the feature nature of the scene is enhanced. According to the coordinate attribute of the feature point, the scene characteristic evaluation index char value is given based on the dispersion degree of the feature point distribution, as shown in Equation (15).
(15)$$char=\sqrt{f_{dif}\left(S_{x}^{2}\right)\times f_{dif}\left(S_{y}^{2}\right)}$$

In Equation (15), $S_{x}^{2}$ is the variance of pixel coordinates x of feature points, $S_{y}^{2}$ is the variance of pixel coordinates y of feature points. The whole equation is the geometric mean of $f_{dif}\left(S_{x}^{2}\right)$ and $f_{dif}\left(S_{y}^{2}\right)$. $f_{dif}$ is a normalization function considering the scene size and image size parameters. Since different image resolutions have different sizes, the variance size of the pixel coordinate distribution is not consistent. In order to unify the description, the variance value needs to be normalized, $f_{dif}$ as shown in Equation (16).
(16)$$f_{dif}=nor\left(C\times \left(1-nor\left(X_{S}\right)\right)\right)$$
(17)nor=atan(x)*2/π

In Equation (16), the function nor is shown in Equation (17), which is a normalized function containing an inverse tangent function that maps variance values greater than 0 to the interval [0,1]. C is the size factor, which reflects the measurement target size and the acquisition image size, and is used to amplify the data difference. $X_{S}$ is the independent variable, i.e., the variance $S_{x}^{2}$ or $S_{y}^{2}$, with expressions as in Equations (18) and (19).
(18)$$S_{x}{}^{2}=\frac{\sum_{i=1}^{n}{\left(x_{i}-\bar{x}\right)}^{2}}{n-1}$$
(19)$$S_{y}{}^{2}=\frac{\sum_{i=1}^{n}{\left(y_{i}-\bar{y}\right)}^{2}}{n-1}$$

After the above derivation, the comprehensive expression of char value can be obtained as shown in Equation (20). The char value is used as the evaluation index of the characteristic of the scene. The value of char describes the dispersion degree of feature points in the x and y directions of pixel coordinates. Its value usually ranges from 0 to 1. The larger the value is, the higher the dispersion of the feature points, and the more discrete and uniformly distributed feature points tend to be independent of each other whilst having better location differentiation.
(20)$$char=\frac{2}{\pi }\sqrt{a\tan\left(C\times \left(1-a\tan\left(\frac{\sum_{i=1}^{n}{\left(x_{i}-\bar{x}\right)}^{2}}{n-1}\right)\times 2/\pi \right)\right)\times a\tan\left(C\times \left(1-a\tan\left(\frac{\sum_{i=1}^{n}{\left(y_{i}-\bar{y}\right)}^{2}}{n-1}\right)\times 2/\pi \right)\right)}$$

So far, without adding additional physical devices, the effective feature points are identified using binocular vision system and the pixel information of the feature points on the left and right eye images are obtained. Then, the location characteristics of these feature information in the scene are enhanced by mathematical methods.

### 4.3. Robust Matching and Tracking Method of Feature Points

#### 4.3.1. Relative Position Relationship Descriptors

The existing feature point detection and matching algorithms do not work well in low-feature scenes precisely because they cannot form this one-to-one mapping relationship. Some people have improved the traditional method to improve the matching accuracy [16]. Some artificial intelligence algorithms build convolutional neural networks, improve descriptors and feature point processing methods, etc. [17,18,19,20,21], which achieves a good matching effect, but also cannot achieve continuous and stable tracking of certain feature points. Their detected feature points are still different in different frames so that the feature point position data cannot be used for the identification of flexible solar wing modalities. In order to solve this problem, this paper proposes the relative positional relationship descriptor in Definition 4, which is described in detail as follows.

First, the region of interest is found, and the distribution statistics of the feature points located in the region of interest are calculated according to the pixel coordinate x direction. Due to the presence of the boundary at the folded connections of the solar wing, the statistical graph of the characteristic pixel coordinate distribution appears as the distributed pixel interval shown in Figure 7. Let us assume that the left boundary of the compartment is $b_{l}$ and the right boundary is $b_{r}$. When Equation (21) is satisfied, the interval is considered to be the boundary of the pixel coordinates, that is, the gap at the junction of the flexible solar wing. $\eta_{boundary}$ in Equation (21) is the threshold for determining the boundary. The boundary feature and the boundary coordinates reflect the position boundary characteristics of the pixel coordinates x direction of the feature points in the region of interest. The camera can completely capture the upper and lower boundaries of the solar wing in the y direction of the pixel coordinate, so the upper and lower boundaries of the solar wing in the image can be used as a marker to locate the feature points.
(21)$$b_{r}-b_{l}>\eta_{boundary}$$

After finding the boundary with positioning function, the relative position relationship descriptor describing the position attribute of feature points is used to describe different feature points, and the feature points are matched. The descriptors of traditional feature point matching algorithms are usually associated with the gray gradient transform of the pixels around the feature points. As shown in Figure 8, the SIFT method, which is the most effective traditional algorithm [16], is used to compare with this method. The SIFT method uses a 4×4×8 vector of 128 dimensions to describe the magnitude and direction of the gray gradient of pixels within a fixed radius of the feature point. However, the similar “appearance” of the feature points in the solar wing scene means that the grayscale variations around the feature points are similar, i.e., the descriptors are similar, and the correct matching relationship cannot be found. In this paper, we describe the distance of each feature point to the x and y boundary using the relative position relationship descriptors based on the found boundary features, i.e., Param1-6 in Figure 8. Since the feature points tend to be neatly arranged according to the queue relation after the projection transformation, the relative position relation descriptors of different feature points have obvious differences. Suppose the pixel coordinate of a feature point is $\left(x_{p}\left(i,j\right),y_{p}\left(i,j\right)\right)$. The i and j in the coordinates describe this queue relationship. The feature point on the right of the current feature point can be described as $\left(x_{p}\left(i+1,j\right),y_{p}\left(i+1,j\right)\right)$. The feature point above the current feature point can be described as $\left(x_{p}\left(i,j+1\right),y_{p}\left(i,j+1\right)\right)$. Given the pixel coordinate x direction threshold $\eta_{p_{1}}$ and y direction threshold $\eta_{p_{2}}$, there is only one feature point satisfying the pixel coordinate limit of Equation (22). At this point, the relative positional relationship descriptor RPD can be expressed in the form of Equation (23).
(22)$$\{\begin{array}{c} x_{p}\left(i,j\right)<b_{r}+\eta_{p_{1}}<x_{p}\left(i+1,j\right)\\ y_{p}\left(i,j\right)<b_{b}+\eta_{p_{2}}<y_{p}\left(i,j+1\right) \end{array}$$
(23)$$RPD=\{\begin{array}{c} desX=x_{p}\left(i,j\right)-b_{r}\\ desY=y_{p}\left(i,j\right)-b_{b} \end{array}$$

In conjunction with the representation of the relative position relationship descriptor in Figure 8, it is clear that Param1 is not equal to Param3, so possible match point 1 can be excluded. When the distance between two feature points is less than a given threshold, the two feature points are considered to be corresponding. In this example, Param1 is approximately equal to Param5, Param2 is approximately equal to Param6, so that the correct matching is obtained.

Compared with traditional methods, the method in this paper can clearly distinguish different feature points with few parameters, which makes feature point matching efficient and accurate.

#### 4.3.2. Evaluation of Matching Effect

To verify the feature point recognition and matching effect of this method, this paper’s method is compared and analyzed with the SIFT method and SuperGlue algorithm, which is based on graph neural network [22,23]. Mikolajczyk et al. [18] used the correct rate and recall rate as the evaluation index of feature matching in their study. Here, the two are fused and the MU value is used as an evaluation metric for comparative analysis. MU is expressed as follows.
(24)$$MU=\frac{1+b^{2}}{\frac{b^{2}}{P}+\frac{1}{R}}$$
where, R is the recall rate of matching results, *P* is the correct rate of matching results, and b is the weight of correct rate in the evaluation index. The values of MU, P, and R range from 0 to 1. When the values of P and R are both 1, it means that all feature points are found and all of them are matched correctly, which is an ideal situation. It should be guaranteed that the location information of feature points on the solar wing could be collected accurately and continuously. We are more concerned about the correctness rate than the recall rate. To meet this requirement, the correctness rate of the matching result is required to be very high, so the value of P needs to be given a high weight. Select b=20.

In order to solve the value of MU, we need to calculate the correct match rate P and the recall rate R, as in Equation (25).
(25)$$\{\begin{array}{c} P=\frac{\mathrm{Number}\;\mathrm{of}\;\mathrm{correctly}\;\mathrm{matched}\;\mathrm{feature}\;\mathrm{points}}{\mathrm{Number}\;\mathrm{of}\;\mathrm{detected}\;\mathrm{feature}\;\mathrm{points}}\\ R=\frac{\mathrm{Number}\;\mathrm{of}\;\mathrm{correctly}\;\mathrm{matched}\;\mathrm{feature}\;\mathrm{points}}{\mathrm{Total}\;\mathrm{number}\;\mathrm{of}\;\mathrm{feature}\;\mathrm{points}} \end{array}$$

The feature points correctly matched in Equation (25) are determined according to the homography constrains. The mapping of the pixel coordinates $p_{1}$ and $p_{2}$ for the left and right eyes of the binocular system, respectively, can be described in the form of Equation (26), where H is the homography matrix. Then, Equation (27) can be obtained by expanding Equation (26), and the theoretical transformation relationship between the right-eye pixel coordinates and the left-eye pixel coordinates shown in Equation (28) can be obtained by further calculation.
(26)$$p_{2}=Hp_{1}$$
(27)$$\left(\begin{array}{c} u_{2}\\ v_{2}\\ 1 \end{array}\right)=\left(\begin{array}{ccc} h_{1} & h_{2} & h_{3}\\ h_{4} & h_{5} & h_{6}\\ h_{7} & h_{8} & h_{9} \end{array}\right)\left(\begin{array}{c} u_{1}\\ v_{1}\\ 1 \end{array}\right)$$
(28)$$\left\{ \begin{array}{l} h_{1}u_{1}+h_{2}v_{1}+h_{3}-h_{7}u_{1}u_{2}-h_{8}v_{1}u_{2}=u_{2}\\ h_{4}u_{1}+h_{5}v_{1}+h_{6}-h_{7}u_{1}v_{2}-h_{8}v_{1}v_{2}=v_{2} \end{array}\right.$$

According to Equation (28), the theoretical pixel coordinate value $\left(u_{2C,}v_{2C}\right)$ of the feature point identified in the left eye in the right eye can be calculated, and the actual value $\left(u_{2T,}v_{2T}\right)$ of the pixel coordinate of the relevant feature point can be read from the right eye image; given the threshold η as shown in Equation (29), when the threshold η is less than the set value, it means that the theoretical calculated value and the actual detected value match, that is, the correct feature point is matched, and then the evaluation index MU value can be found.
(29)$$\eta =\sqrt{{\left(u_{2C}-u_{2T}\right)}^{2}+{\left(v_{2C}-v_{2T}\right)}^{2}}$$

Combined with the evaluation parameter char in Section 4.2, the comprehensive matching parameter (CMP) of the scene can be obtained by taking the summation average of the parameter MU and the parameter char from Equation (30). The value of CMP ranges from 0 to 1, and the closer the value is to 1, the stronger the characteristic of the scene and the better the matching of feature points. Ideally, the value of CMP can be taken to 1, which means that the characteristic of the scene is very high, the features are easy to extract and distinguish, and the matching is completely correct after extracting feature points.
(30)$$CMP=\frac{2}{\frac{1}{char}+\frac{1}{MU}}$$

So far, we have enhanced, identified, and extracted the scene features from two dimensions of the comprehensive matching parameter of the scene. We can extract the continuous pixel coordinate change data of certain feature points steadily under the premise of binocular vision system only. The continuous change data of the spatial 3D position of feature points can be obtained using these feature point pixel coordinates combined with camera parameters. The three-axis velocity of the spatial motion of the feature point in the camera coordinate system is obtained by differentiating the spatial three-dimensional coordinates in the time domain. This velocity information is used as the input parameter of the modal recognition algorithm.

## 5. Modal Identification Algorithms

In Section 4, a continuous sequence of spatial 3D coordinates of feature points in the time domain is obtained by a binocular vision system. The modal parameter information of the system can be obtained by feeding the data into the modal recognition algorithm. ERA and Stochastic Subspace Identification (SSI) are typical representatives of time-domain methods in modal parameter recognition algorithms, which have been successfully applied in specific models in the aerospace field. Based on them, a series of improved algorithms have been born, such as ERA based on data correlation improved by the random decrement (RDT+ERA/DC) technique, and the data-driven stochastic subspace identification (SSI-DATA) technique. The modal omission problem occurs when these two methods are used alone. However, we find that the operations of these two time-domain modal parameter identification methods are not completely independent, but are related. Therefore, in this paper, we consider the fusion of the two methods RDT+ERA/DC and SSI-DATA, aiming to solve their own modal omission problems. The fusion unified calculation flow is shown in Figure 9.
1.The cross-correlation analysis of the vibration response output signal is carried out to obtain the time series of cross-correlation function enhanced by modal information. The cross-correlation function $R_{ijk}(\tau )$
of observation points i and j can be expressed in the form of Equation (31).
(31)$$\begin{array}{l} R_{ijk}(\tau )=\sum_{r=1}^{2n}\sum_{s=1}^{2n}\varphi_{ir}\varphi_{js}a_{kr}a_{ks}\int_{-\infty }^{t}\int_{-\infty }^{t}e^{\lambda_{r}(t+\tau -p)+\lambda_{s}(t-q)}E\left[f_{k}(p)f_{k}(q)\right]dpdq\\ =\sum_{r=1}^{2n}\sum_{s=1}^{2n}\varphi_{ir}\varphi_{js}a_{kr}a_{ks}a_{k}\frac{-e^{\lambda_{r}\tau }}{\lambda_{r}+\lambda_{s}}\\ =\sum_{r=1}^{2n}b_{jr}\varphi_{ir}e^{\lambda_{r}\tau } \end{array}$$
where $a_{k}$ denotes the constant associated with the excitation point k only, $\varphi_{ir}$ and $\varphi_{js}$ denote the r and s order mode oscillations of the observation points i and j, $a_{kr}$ and $a_{ks}$ denote the parameters associated with the excitation point k and the mode order r or s, respectively, $b_{jr}=\sum_{s=1}^{2n}\varphi_{js}a_{kr}a_{ks}a_{k}\frac{-1}{\lambda_{r}+\lambda_{s}}$, $\lambda_{r}$ and $\lambda_{s}$ are the main diagonal elements of the diagonal matrix. If we take f(t) as white noise, we obtain the following:(32)$$E\left[f_{k}(p)f_{k}(q)\right]=a_{k}\delta (p-q)$$
where δ(⋅) denotes the pulse function.

In order to increase the signal-to-noise ratio of the cross-correlation function of the observed signal, the cross-power spectrum of the observed signal is calculated first, and then the cross-correlation function of the observed signal and its time series are obtained by inverse Fourier transform.
2.The generalized Hankel matrix constructed from this time series is shown in Equation (33), and the H(0) and H(1) in the generalized Hankel matrix are extracted.
(33)$$H(t-1)={\left[\begin{array}{cccc} h(t) & h(t+1) & \cdots & h(t+\beta )\\ h(t+1) & h(t+2) & \cdots & h(t+\beta +1)\\ \vdots & \vdots & \ddots & \vdots \\ h(t+\alpha ) & h(t+\alpha +1) & \cdots & h(t+\alpha +\beta ) \end{array}\right]}_{(\alpha +1)l\times (\beta +1)m}$$
where h(t) is the system unit impulse response function at t time, α and β are arbitrary integers.
3.Considering that the CVA weighted processing method is highly resistant to noise, the SVD decomposition of the matrix H(0) after weighting processing is performed to obtain the system observability matrix. The singular value decomposition of H(0) can be obtained as follows:
(34)$$H(0)={\mathrm{USV}}^{T}$$
where U is the (α+1)l×2n dimensional left singular matrix, V is the (β+1)m×2n dimensional right singular matrix, S is the 2n×2n dimensional diagonal matrix. U, V, and S satisfy the requirements of Equation (35).
(35)$$\begin{array}{c} U^{T}U=I_{2n\times 2n}\\ V^{T}V=I_{2n\times 2n}\\ S=\mathrm{diag}(\sigma_{1},\cdots ,\sigma_{2n}) \end{array}$$
4.The least squares method is used to extract the system matrix of the state space model. Introducing $E_{l}^{T}=[\begin{array}{cccc} I_{l} & 0_{l} & \cdots & 0_{l} \end{array}]$, $E_{m}^{T}=[\begin{array}{cccc} I_{m} & 0_{m} & \cdots & 0_{m} \end{array}]$ the system matrix of the system state space model is derived as follows:
(36)$$\left\{ \begin{array}{l} A=S^{-1/2}U^{T}H(1){\mathrm{VS}}^{-1/2}\\ B=S^{1/2}V^{T}E_{m}\\ C=E_{l}^{T}{\mathrm{US}}^{1/2} \end{array}\right.$$
5.The information of the modal parameters is extracted from the system matrix. Based on the solutions of vibration theory and state equation, the i order modal frequency $\omega_{i}$, damping ratio $\xi_{i}$, mode matrix Cψ and amplitude matrix of the system $\psi^{-1}B$ are obtained based on the ERA method as follows:(37){ωi=Re(si)2+Im(si)2ξi=Re(si)ωiCψ=ElTUS1/2ψψ−1B=ψ−1S1/2VTEm
where $s_{i}=\ln(z_{i})/\Delta t$ denotes the eigenvalues of the continuous-time system matrix  A ¯ and Δt denotes the sampling interval.Based on the SSI method, the i order modal frequency $\omega_{i}$, damping ratio $\xi_{i}$, and the vibration matrix of the system Φ are obtained as follows:(38){ωi=|si|ξi=|Re(si)|ωiΦ=Cψ
where $s_{i}=\ln(z_{i})/\Delta t$, Δt denote the sampling time interval.
6.The modal amplitude coherence coefficient (MAC) and modal singular value (MSV) criteria are used to distinguish the real and false modes.

Define the MAC as shown in Equation (39):(39)$$\begin{array}{cc} \gamma_{i}=\frac{\left|{\bar{q}}_{i}^{H}q_{i}\right|}{\left\|{\bar{q}}_{i}\right\|\cdot \left\|q_{i}\right\|}, & i=1,\cdots ,2n \end{array}$$
where the row vector $q_{i}^{H}$ represents the time series of the identified modal amplitude of the i order modal motion, and the value interval of $\gamma_{i}$ is [0,1]. $\gamma_{i}\to 1$ when the i order modal is the true modal, and $\gamma_{i}\to 0$ when the *i* order modal is the false modal.

MSV is a method for determining the degree of contribution of each order of the identified modal to the impulse response signal and is calculated as follows:(40)$$\kappa_{i}=\sqrt{\left|c_{i}\right|(1+\left|z_{i}\right|+\left|z_{i}^{2}\right|+\cdots +\left|z_{i}^{\beta }\right|)\left|b_{i}\right|}$$
where $c_{i}$ is the column vector of the system matrix C and $b_{i}$ is the row vector of the system matrix B. $z_{i}$ is the eigenvalue of the system matrix A. When $\left|z_{i}\right|<1$ and β→μ (μ is a sufficiently large number), Equation (40) becomes the form of Equation (41).
(41)$$\kappa_{i}=\sqrt{\left(\left|c_{i}\right|\cdot \left|b_{i}\right|\right)/\left(1-\left|z_{i}\right|\right)}$$

So far, based on analyzing the difference and connection between ERA and SSI system time domain identification methods, the correlation analysis technique is introduced, the generalized Hankel matrix is constructed, and the SVD decomposition and least squares method are used to solve the system state matrix and extract the modal parameter information. The false and real modes are distinguished according to MAC and MSV. Thus, the fusion of ERA and SSI time domain methods is realized. It effectively solves the problem of modal omission caused by RDT+ERA/DC and SSI-DATA algorithms because the modes with a small vibration response are eliminated as noise.

## 6. Experimental Verification of Modal Identification of Flexible Satellite Appendages

### 6.1. Mathematical Simulation

A three-dimensional model of the whole star which has a central rigid body coupled with flexible attachment is established. The dimension of the central rigid body is set to 600×600×800 mm and the dimension of the flexible solar wing is 1600×400×2 mm, which is the same as the physical model. When the flexible solar wing is subjected to external excitation frequency of 0.66573, the first-order vibration pattern is mainly shown in Figure 10.

### 6.2. Ground Test

#### 6.2.1. Relevant Parameters of the Ground Test

In order to verify the feasibility and measurement accuracy of the method of on-orbit measurement and identification of large flexible structures based on non-contact optical measurement in the low-feature scene of this paper, a flexible solar wing model dynamics test platform is built as shown in Figure 11. The length and width of the three-stage solar array is 1600 mm × 400 mm. The camera baseline length is about 450 mm. The camera resolution is 4096 × 3000. The maximum camera frame rate is 46 fps. Other cameras and lens related parameters are shown in Table 1. During the test, the flexible solar wing is subjected to random external excitation and vibrates freely. The vibration information is collected by the binocular camera, which is then transmitted to the processor for centralized processing.

#### 6.2.2. Results of Ground Tests

Firstly, the binocular vision system is calibrated. The intrinsic parameters of the camera are shown in Table 2. The extrinsic parameters are shown in Equation (42). Then, the superiority of the method is verified in terms of the characteristic of the scene and the uniqueness of the matching. For the scene of this paper, $C=10^{4}$ is selected, and the characteristic of the scene is calculated according to Equation (20), as shown in Table 3.
(42)$$Tform=\left[\begin{array}{cccc} 0.99163 & 0.12630 & -0.02675 & 16.59488\\ -0.11764 & 0.79857 & -0.59029 & -425.66317\\ -0.05319 & 0.58849 & 0.80674 & 138.84437\\ 0 & 0 & 0 & 1 \end{array}\right]$$

The actual effect of the relative position descriptor matching method, SIFT algorithm, and SuperGlue algorithm between different frames is shown in Table 4. From the table, it can be seen that the SIFT algorithm, as the most stable and best performing traditional feature point recognition matching algorithm [16], only has an MU value of 0.5499 in this scene, which indicates that the SIFT algorithm has poor applicability in this scene. The SuperGlue method based on artificial intelligence still maintains a good matching effect in this low-feature scene, but it still has the problem shown in Figure 4. The feature points detected between different frames are quite different, and so the fixed feature points cannot be tracked stably, and the data cannot be used for modal identification. In contrast, the feature point matching algorithm based on the relative position relation descriptor proposed in this paper ensures a better matching effect, and in each frame of the image, the fixed feature points with clear location attributes are tracked stably.

Then, the comprehensive matching parameter of the scene is calculated according to Equation (30), and the results are shown in Table 5. The data show that the comprehensive matching parameter of the SIFT algorithm is only 0.5082, indicating that this algorithm is not suitable for low-feature scenes, which is consistent with its actual performance in scene feature point matching. The matching effect of SuperGlue algorithm has been greatly improved due to the introduction of artificial intelligence and should have a better matching effect if the characteristic of the scene is enhanced. The method used in this paper has the highest CMP value, which not only ensures the correct rate of feature matching but also can stably extract the continuous position changes of the required feature points in each frame, which has better practical significance.

Finally, the spatial 3D coordinates of the target measurement point were tracked using the calibrated binocular camera. The 3-axis spatial position and velocity of the feature points in the camera coordinate system were obtained as shown in Figure 12 and Figure 13. To evaluate the measurement accuracy of the vision system in this experimental scheme, five OptiTrack series Prime^X^22 cameras were used to form a high-precision camera set with a frame rate of 360 fps and a 3D measurement accuracy of ±0.15 mm. The targets used by OptiTrack cameras were pasted at the three target measurement points, and the 3D spatial positions of the three target points were measured. Due to the influence of the target’s height, the coordinates of the spatial points obtained by the OptiTrack camera set and the camera in this experimental scheme are not exactly the same. To solve this problem, the measured data from the OptiTrack camera set were transformed to the camera coordinate system of this experimental binocular camera. Then, the Y coordinate difference of the adjacent measured points in the camera coordinate system was calculated. This value was compared with the Y coordinate difference of the feature points measured by the binocular camera in this paper’s experimental scheme. The absolute error curve of measurement was obtained as shown in Figure 14. The dashed line in Figure 14 shows the average absolute error in 102 sampling frames, and the value is 0.937044118 mm. Hereby, the binocular camera scheme in this paper can achieve sub-millimeter level measurement accuracy, which is an effective on-orbit measurement method.

After obtaining the motion information of the free vibration process of the flexible solar array under external excitation, the modal identification is carried out by using the fusion method of RDT+ERA/DC and SSI-DATA. The results are shown in Table 6 and Figure 15. The true value of the vibration frequency of the flexible solar wing is 0.7146 Hz, so the experimental identification error is 2.82% and the mathematical analysis error is 6.84%.

## 7. Conclusions

This paper proposes a non-contact optical measurement method for the on-orbit measurement and identification of the modal parameters of large flexible structures of satellites, and the following conclusions are obtained.

The concept of the comprehensive matching parameter of scenes is proposed, based on which the features of the scenes themselves are enhanced through theoretical derivation and mathematical transformation. The relative position relationship descriptor is proposed for the description and matching of feature points so that stable tracking measurements of specific feature points can be achieved by using only binocular cameras without relying on physical identification points.

To demonstrate the feasibility of the above scheme, a physical experimental platform is built for experimental testing and the binocular measurement results are evaluated. The average absolute measurement error is about 0.937 mm. Compared with the traditional targeting scheme, the scheme in this paper takes into account the measurement accuracy while simplifying the experimental setup and improving the stability of the system.

In this paper, the binocular vision system measurement data are applied to the modal identification method based on the fusion of Eigensystem Realization Algorithm and Stochastic Subspace Identification. The first-order frequency of the flexible solar wing vibration is 0.7364 Hz and the discrimination error is 2.82%, which proves that the non-contact optical measurement scheme has good modal identification accuracy and practical application value. In the future, we can consider combining the feature enhancement method or the feature point description method proposed in this paper with the neural network to improve feature point detection efficiency and recognition accuracy.

## Figures and Tables

**Figure 1 sensors-23-01878-f001:**
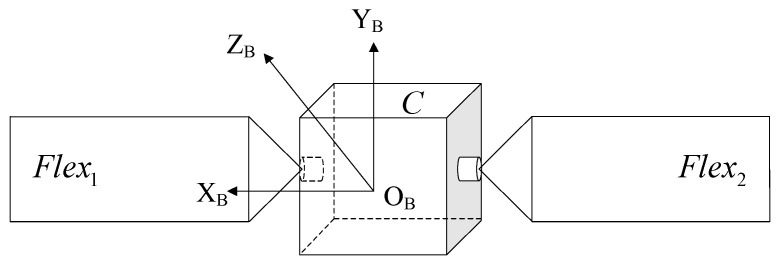
Whole star model and coordinate system.

**Figure 2 sensors-23-01878-f002:**
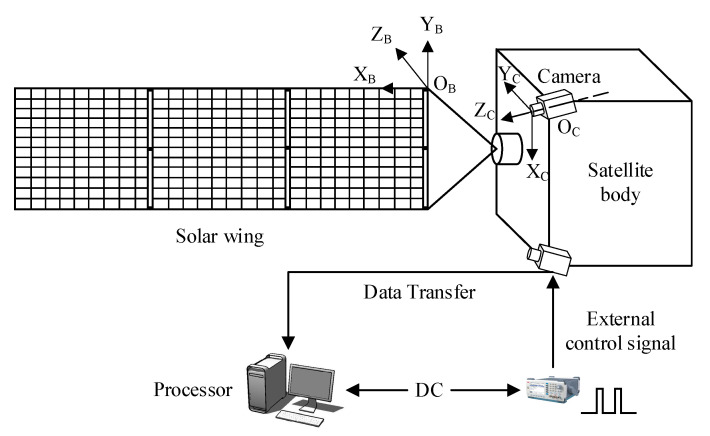
On-orbit measurement simulation test program diagram.

**Figure 3 sensors-23-01878-f003:**
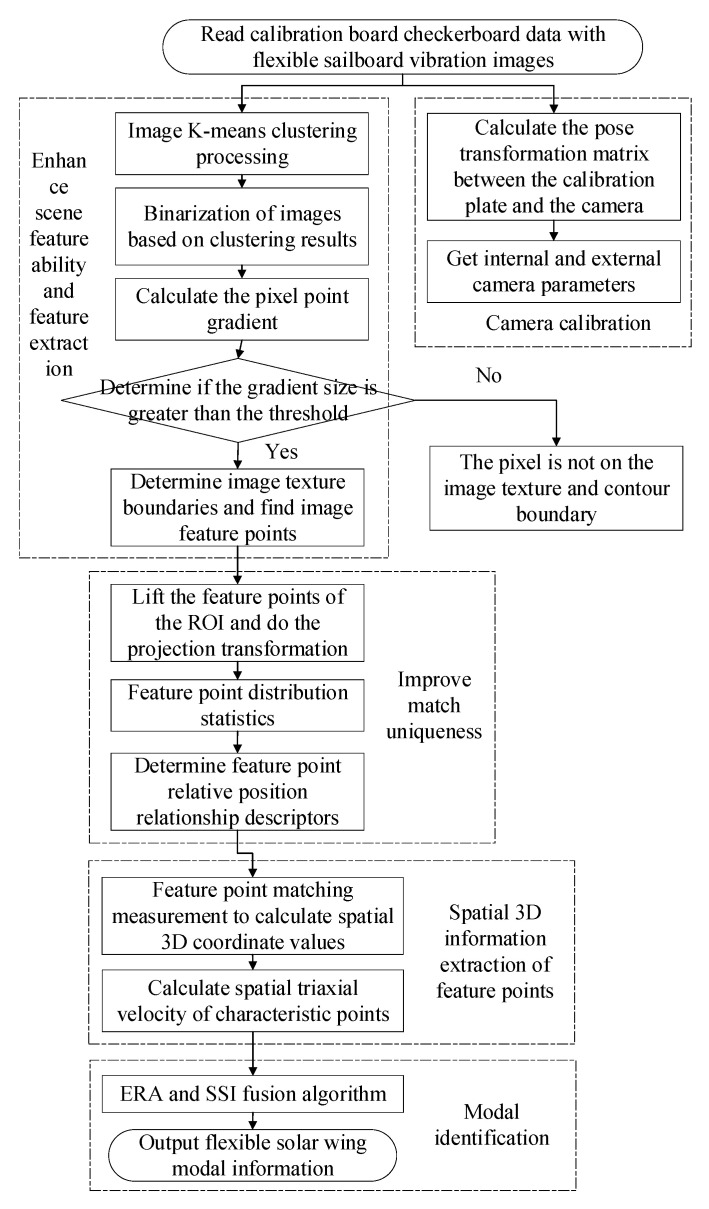
Flow chart of the test program.

**Figure 4 sensors-23-01878-f004:**
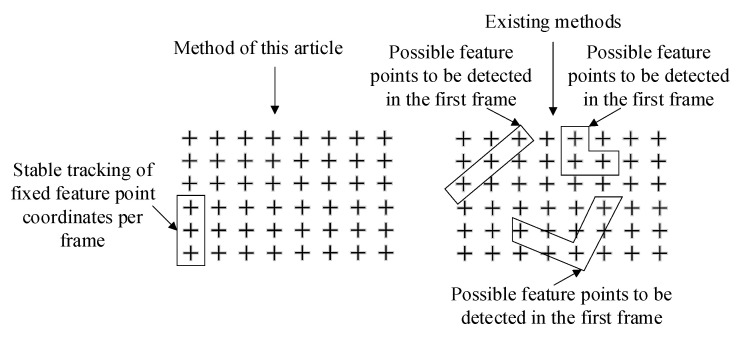
Schematic diagram of feature point detection between different frames.

**Figure 5 sensors-23-01878-f005:**
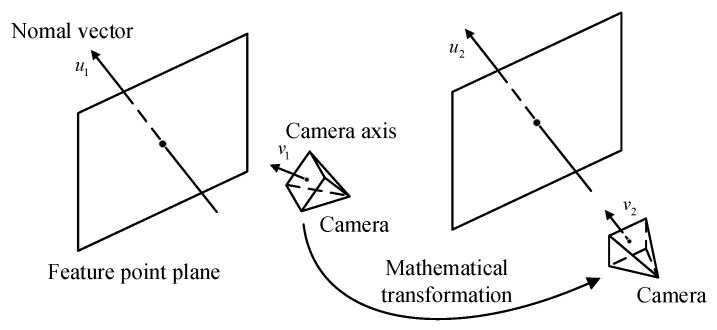
Changes in camera view angle.

**Figure 6 sensors-23-01878-f006:**
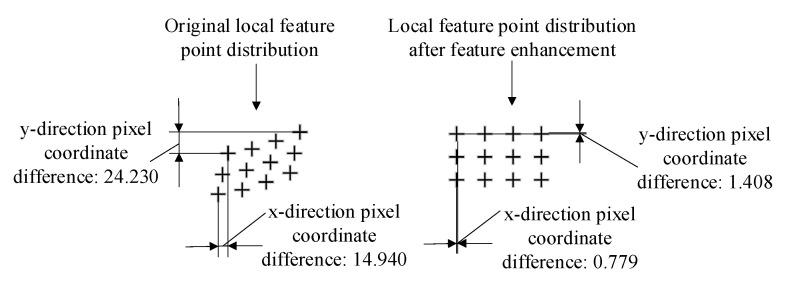
Comparison before and after scene characteristic enhancement. (Symbols + in the image are the locations of feature points).

**Figure 7 sensors-23-01878-f007:**
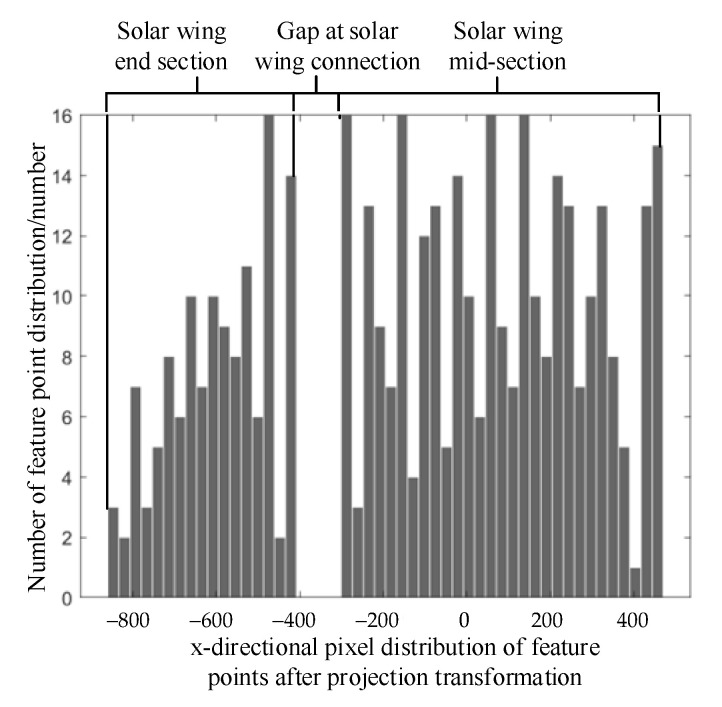
Statistical map of feature point distribution.

**Figure 8 sensors-23-01878-f008:**
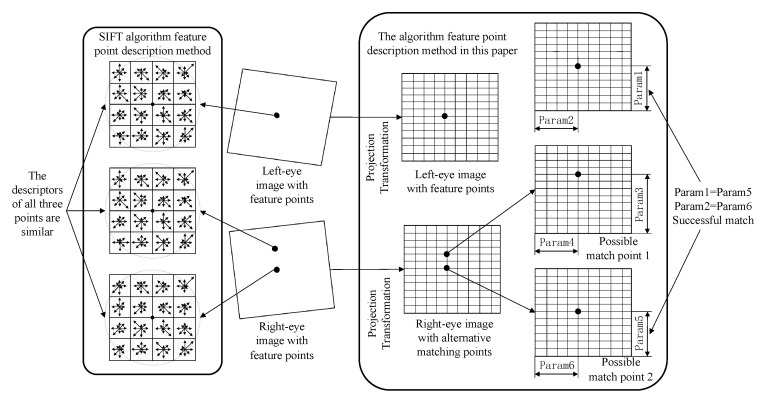
Comparison of feature point description methods.

**Figure 9 sensors-23-01878-f009:**
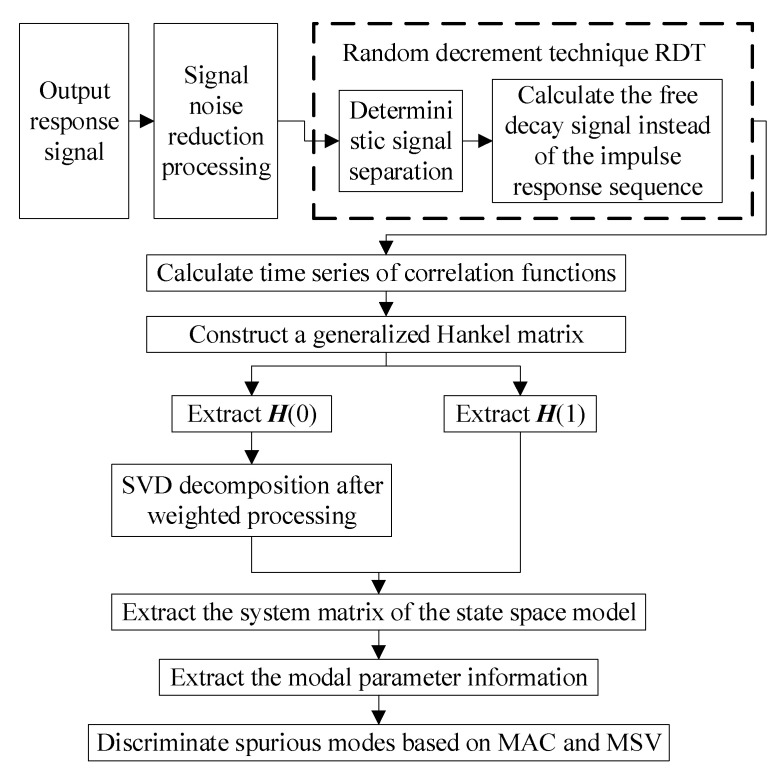
Flow chart of modal recognition fusion algorithm.

**Figure 10 sensors-23-01878-f010:**
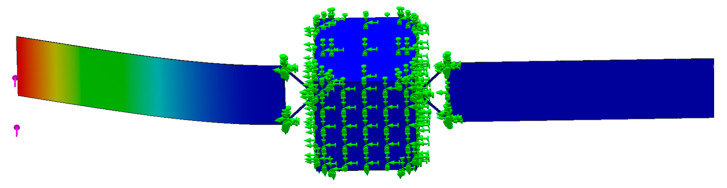
Whole star flexible solar wing frequency analysis. (The green symbol in the figure represents the constraint, and the pink arrow represents the external disturbance.)

**Figure 11 sensors-23-01878-f011:**
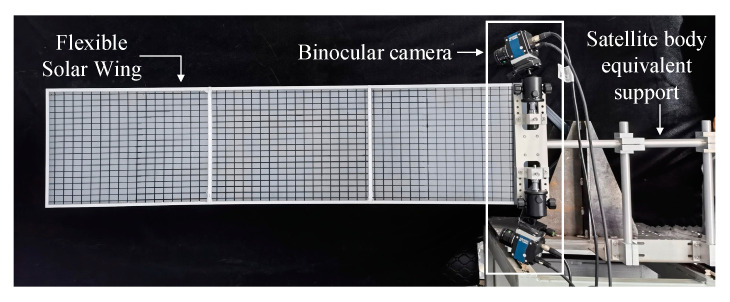
Physics experiment platform.

**Figure 12 sensors-23-01878-f012:**
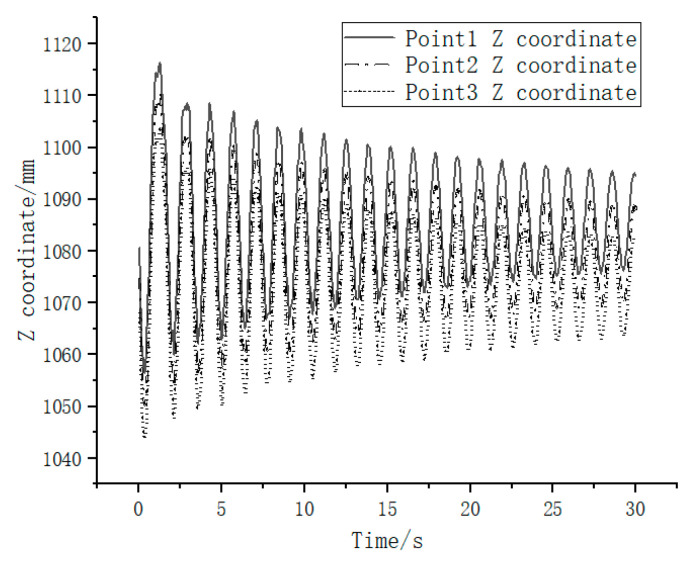
Measurement points Z-directional position versus time.

**Figure 13 sensors-23-01878-f013:**
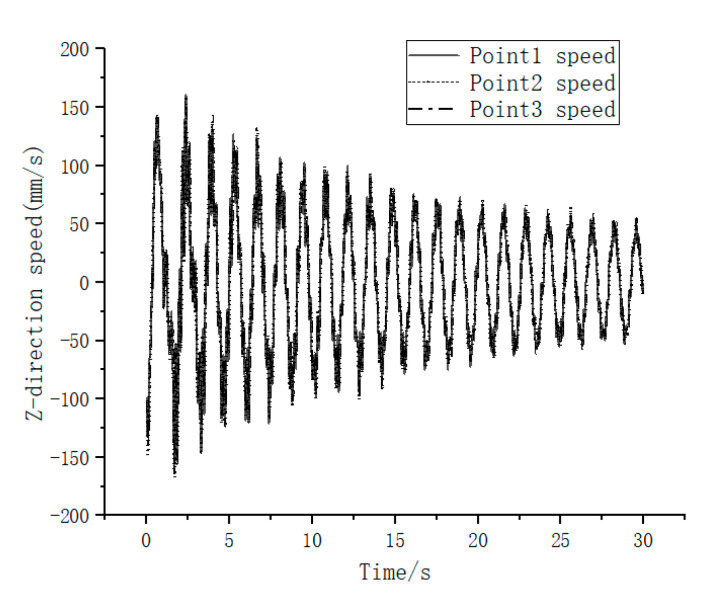
Measurement points Z-directional velocity versus time.

**Figure 14 sensors-23-01878-f014:**
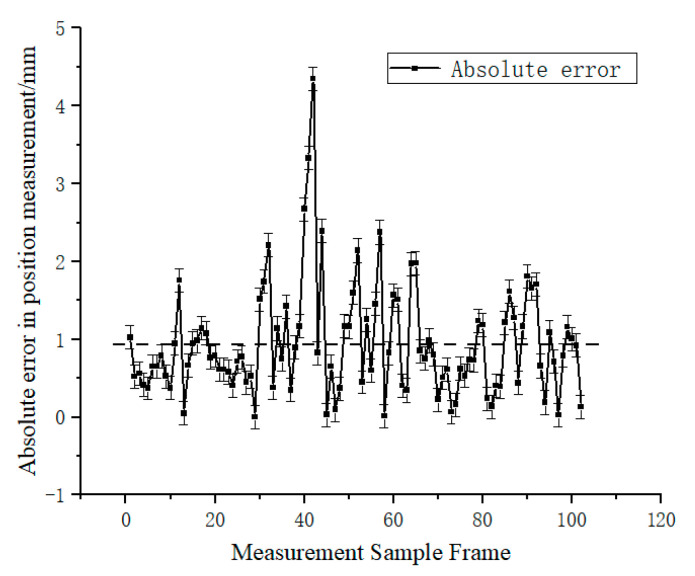
Measurement error of this experimental vision scheme.

**Figure 15 sensors-23-01878-f015:**
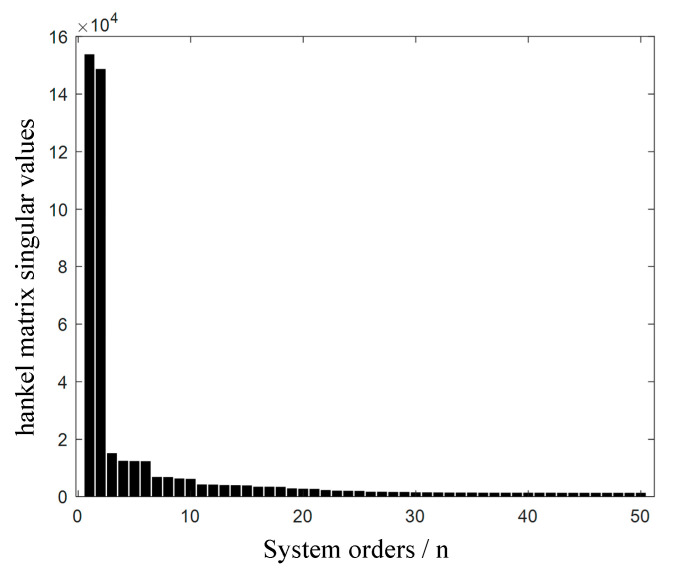
Modal parameter identification.

**Table 1 sensors-23-01878-t001:** Camera and lens parameters table.

Camera Pixel Size	Camera Sensor	Spectrum	Exposure Time	Lens Focal Length f	Lens Aperture F
3.45 μm× 3.45 μm	1.1 inches, Global Shutter	Mono	3 ms	16 mm	6

**Table 2 sensors-23-01878-t002:** Binocular camera internal parameters.

Camera	$f_{x}$	$f_{y}$	$c_{x}$	$c_{y}$
Left eye	4703.986588930	4703.111975069	2079.232347048	1475.465152926
Right eye	4725.708015617	4723.713940567	2052.554124594	1490.440881834

**Table 3 sensors-23-01878-t003:** Scene Characteristic Comparison.

	Before Enhanced Features	After Enhanced Features
Scene characterization char	0.4665	0.9091

**Table 4 sensors-23-01878-t004:** Comparison of feature point matching results of different algorithms.

	Total Points	Detected Points	Correct Matching Points	MU
SIFT algorithm	306	14	8	0.5499
SuperGlue algorithm	306	18	16	0.8631
Methods in this paper	306	3	3	0.8389

**Table 5 sensors-23-01878-t005:** Comparison of the comprehensive matching parameter of scenes.

	char	MU	CMP
SIFT algorithm	0.4665	0.5499	0.5048
SuperGlue algorithm	0.4665	0.8631	0.6057
Methods in this paper	0.9091	0.8389	0.8726

**Table 6 sensors-23-01878-t006:** Modal recognition results.

Order	Theoretical Value/ Hz	Identification Value/ Hz	Error
1, 2	0.7364 (High precision solutions)	0.73478 (Experimental protocol of this paper)	2.82% (Experimental protocol of this paper)

## Data Availability

Not applicable.

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
