# Peer review of "Non-Contact Measurement and Identification Method of Large Flexible Space Structures in Low Characteristic Scenes"

_sensors, 2023, doi:10.3390/s23041878_

Round 1

Reviewer 1 Report

Proposed article deals with the subject of contact-less optical measurement method regarding the parameters of flexible structures of satellites by means of binocular cameras. Proposed method gives valuable results and proves to be useful in future applications. Even though the proposed article has 24 pages, it's structure and style of writing  allow for easy tracking of the information.

On page 4-Lines 147-153 should be deleted (part of the template remained). Besides that, I recommend accepting the article.

Author Response

Dear Editor:

Thank you very much for reviewing this article. Also thank you for your recognition of our work. The changes to your comments are in the attachment.

Tianming Cheng

Reviewer 2 Report

The authors present an work where it is used stereovision to collect vibration information from an artificial satellite wing panel. 

I found the manuscript hard to read, and not very well organized. The manuscript is a bit long and vague. For example, you describe very well the calibration method, but is something well known that could be described in annexed files. 

You write "definition 1,2, etc. along the manuscript, but it is not clear what you are defining. Under 4.3.1 what is the "definition"? are you defining the title of the sub-chapter. 

Also, I barely notice any difference (I mean graphical) between 4.3 XXXX and 4.3.1 XXXX 

In the begenning of chapter 3:

" The Materials and Methods should be described with sufficient details to allow others to replicate and build on the published results. Please note that the publication of your manuscript implicates that you must make all materials, data, computer code, and protocols associated with the publication available to readers. Please disclose at the submission stage any restrictions on the availability of materials or information. New methods and protocols should be described in detail while well-established methods can be briefly described and appropriately cited. "

Is this part of the manuscript? 

From the scientific side, on the ground tests, you compared the results with another camera as a ground-truth (optiTrack). But it is not clear how do you guarantee that OptiTrack is giving enough accurate results? Couldn't you use another method?  

Therefore, considering the journal "sensors" is a leading journal in the area, I recommend major revisions, trim the paper, and make it more readable.

Author Response

(The authors gave the same response as above.)

Reviewer 3 Report

The paper presents an innovative approach for the non-contact measurement and identification of large, flexible space structures in low characteristic scenes. The authors highlight the importance of satellite attitude control accuracy and the impact of the vibration of the satellite's flexural structure on this accuracy. They argue that a stable and reliable vibration frequency identification method is needed.

The authors propose a new concept of "The comprehensive matching parameter" of scenes to describe the characteristics of non-contact optical measurement. They also provide a detailed explanation of the basic connotation and evaluation index of this concept. The proposed method improves recognition accuracy and matching uniqueness of features through equivalent spatial transformation and novel relative position relationship descriptor. This is a good work as it solves existing problems in non-contact measurement technology without the need for additional hardware devices.

The authors also demonstrate the effectiveness and superiority of the proposed method through mathematical simulation and ground testing. The results of the simulation and testing support the validity of the proposed method. 

1. Writing: The authors could benefit from more precise and concise language. This would help to improve the clarity of the paper and make it more accessible to a wider audience. The authors should consider having a native English speaker review the paper for grammar, spelling and some punctuation errors. This will help to improve the overall readability and clarity of the paper.

2. Limitations and future directions: The authors could provide a analysis of the limitations of the proposed method and discuss potential areas for improvement. They could also consider discussing the generalizability of the method to other types of structures or applications. Additionally, they could suggest possible future research directions that could build on the work presented in the paper. This will help to give context to the significance of the proposed method, and provide a foundation for future research in this field.

Author Response

(The authors gave the same response as above.)

Round 2

Reviewer 2 Report

The authors greatly improved the readability of the manuscript. Therefore, I recommend to publish in the present form.